# Psychosocial Impact of the COVID-19 Pandemic in Brazilian Post-Peak Period: Differences Between Individuals with and Without Pre-Existing Psychiatric Conditions

**DOI:** 10.3390/ijerph22010027

**Published:** 2024-12-29

**Authors:** Rodrigo Sanches Peres, Pedro Afonso Cortez

**Affiliations:** Instituto de Psicologia, Universidade Federal de Uberlândia, Avenida Pará, 1720-Bloco 2C, Campus Umuarama, Uberlândia 38405-240, MG, Brazil; rodrigosanchesperes@ufu.br

**Keywords:** mental health, pandemics, COVID-19, public health

## Abstract

(1) Background: Validated instruments to measure mental health variables related to sanitary crises can provide data for prevention or intervention plans. The objectives of this study were: (1) to evidence the psychometric factorial internal structure of the Battery for Assessing Mental Health–Pandemic Version (BASM-P) in the sample; (2) to investigate the psychosocial impact of the COVID-19 pandemic in Brazilian post-peak period among individuals with and without pre-existing psychiatric conditions using the BASM-P; and (3) to analyze relationships between the mental health variables measured by the BASM-P in both groups. (2) Methods: This is an internet-based quantitative, cross-sectional study with a non-probabilistic convenience sample. The participants (*n* = 209) were divided into a non-psychiatric group (*n* = 168) and a psychiatric group (*n* = 41). The instruments were the BASM-P and a sociodemographic questionnaire. Data were collected throughout the second semester of 2022 and analyzed with JASP software. (3) Results: The BASM-P demonstrated robust psychometric factorial internal structure. Significant differences were observed between the two groups across all variables. In the non-psychiatric group, obsessive thoughts presented strong connections to fear, distress, and grief from job loss. (4) Conclusions: This study highlights the central role of obsessive thoughts in shaping the psychosocial impact of the COVID-19 pandemic in the Brazilian post-peak period.

## 1. Introduction

The international spread of infectious diseases tends to represent a major challenge for mental healthcare. This became clear with the Severe Acute Respiratory Syndrome (SARS) and the Ebola Virus Disease (EVD) outbreaks in past decades when a myriad of psychosocial consequences was observed in both psychiatric and non-psychiatric populations, according to several researchers [1,2,3,4]. Validated instruments to measure mental health variables related to sanitary crises like these, such as scales and questionnaires, are essential to address the aforementioned challenge, as they can provide data for appropriate prevention or intervention plans definition, health services reorganization, human and financial resources reallocation, and public policies improvement, for example [5,6].

Based on this assumption, a review published in the initial phase of the COVID-19 pandemic presented a selection of instruments potentially useful to determine the presence and the severity of a broad range of emotional and behavioral problems associated with this then-new urgent health-threatening situation [7]. This review led a group of researchers to compose a task force aimed at adaptation and validation, to Brazil, of instruments for mental health variables screening in pandemics. It was a relevant initiative, since a few months after the first confirmed case in Brazil, this country became an epicenter of the COVID-19 pandemic [8], consequently worsening pre-existing health, political, economic, and social problems [9].

Focusing on semantic and linguistic aspects [10], the Brazilian task force adapted and validated the following instruments: COVID-19 Phobia Scale [11], COVID-19 Stress Scale [12], Coronavirus Anxiety Scale [13], Questionnaire on Perception of Threat from COVID-19 [14], Fear of COVID-19 Scale [15], COVID-19 Peritraumatic Distress Index [16], Obsession with COVID-19 Scale [17], Traumatic Grief Inventory Self-Report Version [18], Job Loss Grief Scale [19], and Adaptation to Change Questionnaire [20]. The pertinence of this clarification lies in the fact that this group of researchers also developed and validated the Battery for Assessing Mental Health–Pandemic Version (BASM-P), an original instrument that, as will be detailed later, covers main mental health variables evaluated by the aforementioned instruments [21].

Therefore, the BASM-P contemplates a spectrum of emotional and behavioral problems manifested in terms of phobia, stress, anxiety, grief, coping strategies, among others, and this is a practical and economical advantage. In addition, the BASM-P does not require specialized training, since it is easily understandable, applicable, and interpretable. Last, but not least, the BASM-P is not restricted to the measurement of COVID-19 psychosocial impact, so can be utilized in future pandemics or sanitary crises, which are considered inevitable [22]. Consequently, it can rapidly enable access to relevant data for the immediate mental health systems’ strengthening, as is necessary for urgent health-threatening situations, mainly in low- and middle-income countries [23,24].

To provide an overview of the psychosocial impact of the COVID-19 pandemic in Brazil, it is interesting to highlight the findings of some research on mental health variables similar to those measured by BASMP-P. One of these studies [25] found predominantly mild or severe levels of peritraumatic distress (71.3%) and mild or severe levels of fear (54.2%) in a Brazilian sample evaluated in April 2020. Also, in the initial phase of the COVID-19 pandemic in Brazil, Peres et al. [15] and Meller et al. [26] identified that higher levels of fear were associated with symptoms of anxiety, stress, depression, and suicidal ideation. Convergent results were reported in a Brazilian study centered on two subscales of the Mental Impact and Distress Scale since they revealed a positive correlation between behavioral and/or cognitive distress expressions, symptoms of anxiety and depression, and maladaptive responses [27].

A high prevalence of avoidance (59.2%), intrusion (46.8%), and hyperarousal (50.1%) were observed as manifestations of the early psychological impact of the COVID-19 pandemic by Campos et al. [28] in a Brazilian research. It should be noted that the psychiatric patients included in the respective sample presented an increased risk of developing these manifestations, and, consequently, detrimental coping strategies and adjustment problems. According to another research developed in Brazil [18], specifically in the first semester of 2021, traumatic grief indicators were positively correlated with obsessive thoughts [17]. By the way, higher scores of obsessive thoughts about COVID-19 were associated with higher rates of negative affects [29] and with poor quality of life [30], according to research from the “country of soccer”. In contrast, Cortez et al. [20] observed, in a Brazilian sample, a positive correlation between adaptative responses and good quality of life.

Given the above considerations, the objectives of this study were: (1) to evidence the psychometric factorial internal structure of BASM-P in the sample; (2) to investigate the psychosocial impact of the COVID-19 pandemic in Brazilian post-peak period among individuals with and without pre-existing psychiatric conditions using the BASM-P; and (3) to analyze relationships between the mental health variables measured by the BASM-P in both groups. We proposed the following hypotheses: (1) the psychiatric group will present higher levels of distress, obsessive thoughts, and maladaptive coping compared to the non-psychiatric group, as measured by the BASM-P; (2) the centrality of obsessive thoughts and distress will be greater in psychiatric patients, reflecting a higher complexity in the connections between mental health variables; and (3) maladaptive coping will have a stronger association with stress in the psychiatric group than in the non-psychiatric group.

To justify this study, it should be emphasized that the utilization of validated instruments to measure mental health variables related to sanitary crises can contribute to the Brazilian public health system—named Sistema Único de Saúde (SUS)—providing data to accommodate both psychiatric and non-psychiatric populations needs in terms of mental health. On the one hand, these data could guide the tracking of individuals with subclinical symptoms, the monitoring of mental disorders determinants, and the promotion of preventive actions at the primary care level [27]. A short and easy-to-use instrument such as the BASM-P is particularly suitable for this purpose since most Brazilian primary care teams do not count on professionals trained for addressing mental health problems [31].

On the other hand, a tendency towards worsening pre-existing psychiatric conditions was observed after the beginning of the COVID-19 pandemic in Brazil, in part because it promoted the accentuation of social inequalities in this middle-income country [32]. The evaluation of mental health variables in this scenario is crucial to enhance access to specialized care, available at the secondary and tertiary care levels [31]. It is imperative to warn that many mental health services in Brazil present accessibility issues, a problem to be overcome in order to minimize the impact of mental disorders on individuals and families, and also their direct and indirect economic and social costs [33].

## 2. Materials and Methods

### 2.1. Study Design

This is an internet-based quantitative, cross-sectional study with a non-probabilistic convenience sample.

### 2.2. Participants

This study included a total of 209 participants, divided into a non-psychiatric group (*n* = 168) and a psychiatric group (*n* = 41). The inclusion criteria were: (1) ≥18 years; (2) read, write, and understand Portuguese; and (3) provide informed consent for processing the respective clinical records and for participation in this study. For the non-psychiatric group, participants were included if they had no history of mental disorders and/or psychopharmacological treatment or psychotherapy during the COVID-19 according to their respective clinical records. This group consisted of individuals being assisted by the Family Health Strategy, a community-based initiative responsible for comprehensive and longitudinal health care provision in the SUS context, and which is oriented towards the production of citizenship [34].

Individuals in the psychiatric group were included if they had a pre-existing mental disorder and had been undergoing psychopharmacological treatment and/or psychotherapy follow-up for at least six months during the COVID-19 pandemic, according to their clinical records. Formal psychiatric diagnoses were not obtained for this study due to ethical restrictions within the health services in which participants in the psychiatric group were being assisted. Instead, inclusion was based on treatment history, ensuring the separation between the two groups in this study while respecting ethical considerations related to diagnostic disclosure.

The average age of the non-psychiatric group was 37.601 years (SD = 12.840), ranging from 18 to 67 years. The psychiatric patients had an average age of 34.902 years (SD = 13.755), with ages ranging from 18 to 68 years. Among the non-psychiatric group, there were 79.17% white participants, while the psychiatric group included 68.29% white participants. Most participants were women: 72.62% in the non-psychiatric group and 78.05% in the psychiatric group. Regarding educational attainment, there were 86.90% of participants with a degree in the non-psychiatric group, compared to 90.24% with a degree in the psychiatric group.

Regarding COVID-19 diagnosis, 58.93% of participants from the non-psychiatric group had no diagnosis. Among the psychiatric patients, 63.41% of participants had no diagnosis of COVID-19. A total of 73.81% of participants from the non-psychiatric group and 80.49% of participants from the psychiatric group did not experience a wage reduction. In terms of grief experienced during the pandemic, 71.43% of participants from the non-psychiatric group did not report grief in the family. Among the psychiatric patients, 58.54% did not experience family grief. This comprehensive description indicates that the groups are homogeneous in terms of sociodemographic variables, as shown in Table 1.

### 2.3. Setting

The present study was developed in Brazil, the largest country in Latin America, with a territory of 8.5 million km^2^, and the most populous nation in this region, since it had 212.6 million inhabitants in July 2024 [35]. More specifically, the present study was carried out in the Brazilian Southeast macro region, which is made up of the states of Rio de Janeiro, São Paulo, Espírito Santo, and Minas Gerais, and is the major powerhouse of the country’s economy. It is relevant to inform that Brazil counts on a complex public health system, on which around 75% of the Brazilian population is totally dependent [36]. Created in 1990, the SUS is composed of health services provided by federal, state, and municipal governments and is focused on the primary care level [31]. The Brazilian mental health policy integrates the SUS, was implemented in 2001, and is oriented by a model of psychosocial care convergent with the principle of deinstitutionalization [33]. However, the SUS was not able to appropriately address the physical and mental health needs of the Brazilian population during most of the COVID-19 pandemic, mainly because political issues convoluted the implementation of the public health measures recommended to contain the COVID-19 properly [32].

### 2.4. Instruments

#### 2.4.1. Battery for Assessing Mental Health–Pandemic Version (BASM-P)

It is a 30-item self-report instrument designed to evaluate the psychosocial impact of pandemics or sanitary crises. The BASM-P covers varied mental health variables since it includes ten factors: (1) phobia; (2) stress; (3) anxiety; (4) perception of vulnerability to illness; (5) fear; (6) distress; (7) obsessive thoughts; (8) traumatic grief; (9) grief from job loss; and (10) maladaptive coping. Each factor is assessed by three items, providing a score that reflects the degree of emotional and behavioral problems related to that specific factor. The overall score indicates general distress due to an ongoing pandemic or sanitary crisis period. The response scale is a five-point Likert scale ranging from “1 = Never” to “5 = Always” [21].

The content validity of the BASM-P was established through evaluation by a committee of experts, ensuring the relevance and clarity of the items. A semantic analysis was conducted with the target population to confirm item comprehensibility. Internal structure validity was supported by factor analysis, with a satisfactory fit. The confirmatory factor analysis proposed a model with ten specific factors and a second-order general factor, demonstrating robust fit indices. Reliability was confirmed with high internal consistency for all factors, with Cronbach’s Alpha and McDonald’s Omega values indicating strong reliability (α = 0.91; ω = 0.92).

The BASM-P is intended for use in both clinical and research settings in a dimensional perspective. It adopts a dimensional approach, assessing mental health on a spectrum rather than using categorical cut-off points, allowing for a more nuanced evaluation. It provides valuable insights into specific areas of psychosocial impact, enabling targeted strategies. The instrument can be administered online or in person, is suitable for use with diverse populations, and has a specific manual available [21].

#### 2.4.2. Sociodemographic Instrument

Sociodemographic data were collected using a structured questionnaire designed to capture a range of personal information from the participants. This instrument included items related to: (1) age; (2) gender; (3) educational attainment; (4) ethnicity; (5) COVID-19 diagnosis; (6) wage reduction; and (7) experiences of grief during the pandemic. Participants reported their current age, which was used to calculate the mean and range for both non-psychiatric and psychiatric groups. Gender was identified as either male or female. Educational attainment was categorized as “Not Graduated” for those without a degree and “Graduated” for those with a degree.

Based on Brazilian self-report criteria, participants identified themselves as either white or non-white. This self-report criterion focuses on personal comprehension of skin color, which may differ from other cultural understandings of ethnicity. Participants also indicated whether they had been diagnosed with COVID-19 (Yes/No), if they had experienced a reduction in their salary (Yes/No), and if they had experienced grief due to the loss of a family member during the pandemic (Yes/No). The data collected through this sociodemographic questionnaire provided a comprehensive overview of the participants’ backgrounds and facilitated the comparison of non-psychiatric and psychiatric groups on various pandemic-related factors.

### 2.5. Data Collection

Data were collected throughout the second semester of 2022 using an online survey form on the Survey Monkey platform for eligible participants, with their prior consent. It is important to note that on 3 June 2022, 86.6% of the Brazilian population had received only the first immunizing dose for COVID-19, since, due to negligence on the part of President Bolsonaro’s government, the vaccination campaign was intensified belatedly on the second quarter of 2021 [37]. During the data collection, no lockdowns were in place, and non-pharmacological preventive measures as the use of face masks were not emphatically encouraged by the Ministry of Health. According to the World Health Organization’s classification of pandemic phases [38], it is possible to state that Brazil was in the post-peak period in the second semester of 2022, because, after the third wave of contamination, caused by the Omicron variant, the country was recording the stabilization of the incidence rate and the decrease in the mortality rate. However, part of the population was acting as if the COVID-19 pandemic was already over, and the Bolsonaro effect could explain this [39].

### 2.6. Ethical Procedures

This study obtained approval from the Brazilian Ethical Committee. All participants provided informed consent for the use of their data, which were treated in aggregate form for public health purposes.

### 2.7. Data Analysis

Data analysis was conducted using JASP software version 0.18.3.0. Descriptive statistics and chi-square tests were employed to analyze participants’ general information. Effect sizes were evaluated using Cohen’s d (up to 0.20 = small; up to 0.50 = medium; above 0.80 = large) and η^2^ (up to 0.01 = small; up to 0.06 = medium; above 0.14 = large). Factor analysis was performed to assess the dimensionality of BASM-P. Cronbach’s Alpha and McDonald’s Omega were calculated to evaluate the reliability of the factors. *t*-tests were conducted to examine differences between psychiatric and non-psychiatric groups on the BASM-P factors. Additionally, network analysis was used to describe processual differences in the BASM-P factors between the groups.

## 3. Results

### 3.1. Psychometrics Factorial Internal Structure

The Kaiser-Meyer-Olkin (KMO) test was conducted to evaluate the adequacy of the sample—which was homogeneous in sociodemographic terms, it is worth reinforcing—for factor analysis. The overall KMO value was 0.882, indicating a meritorious level of sampling adequacy. Bartlett’s test of sphericity was significant, *χ*^2^(45) = 892.089, *p* < 0.001, indicating that the correlations between items were sufficient for factor analysis. A confirmatory factor analysis was conducted using promax rotation based on the best indicator for each content from the previous study [21]. The analysis revealed a single-factor solution fit, accounting for 43.8% of the variance. Factor loadings were reasonable and are presented in Table 2.

The overall fit of the model was evaluated using several indices. The chi-squared test for model fit was significant (*χ*^2^(35) = 90.119, *p* < 0.001). Additional fit indices were as follows: Root Mean Square Error of Approximation (RMSEA) = 0.087 (90% Confidence Interval–CI [0.065, 0.109]), Standardized Root Mean Squared Residual (SRMR) = 0.044, Tucker-Lewis Index (TLI) = 0.916, Comparative Fit Index (CFI) = 0.935, and Bayesian Information Criterion (BIC) = −96.863. These indices indicate an acceptable fit for the unidimensional model. The internal consistency of the 10-item measure was assessed using McDonald’s omega. The point estimate for McDonald’s omega was 0.871, indicating excellent internal consistency. The 95% confidence interval for McDonald’s omega ranged from 0.844 to 0.897, further supporting the reliability of the measure. The mean score was 1.974 with a standard deviation of 0.791 (Min = 1; Max = 5).

### 3.2. Differences Between Psychiatric and Non-Psychiatric Groups on BASM-P

Descriptive statistics for the BASM-P General Factor scores were calculated for both groups. The non-psychiatric group (*n* = 168) had a mean score of 1.946 (SD = 0.711, SE = 0.055), with a coefficient of variation of 0.365. The psychiatric group (*n* = 41) had a higher mean score of 2.741 (SD = 0.887, SE = 0.139), with a coefficient of variation of 0.324. These descriptive statistics indicate that the psychiatric group had higher and more variable scores on the BASM-P General Factor compared to the non-psychiatric group. The non-psychiatric group showed a more concentrated distribution of scores around the mean, with fewer extreme values. The boxplot indicated a lower median score and a narrower interquartile range (IQR). The psychiatric group displayed a more spread-out distribution, indicating greater variability in scores. The boxplot showed a higher median score and a wider IQR, with individual scores spread out across a wider range. A raincloud plot was generated to visually compare the BASM-P General Factor scores between the non-psychiatric and psychiatric groups in Figure 1.

An independent samples *t*-test was conducted to compare the BASM-P General Factor scores between the non-psychiatric (0) and psychiatric (1) groups. The results indicated a significant difference in scores between the two groups, t(206) = −6.101, *p* < 0.001. The effect size, measured by Cohen’s d, was −1.063 with a standard error (SE) of 0.184. Additionally, the Vovk–Sellke Maximum *p*-Ratio (VS-MPR) suggested very strong evidence against the null hypothesis, reinforcing the differences between the groups. A Mann–Whitney U test was also performed to confirm these findings. The results were consistent, with U = 1676.500, *p* < 0.001, and an effect size (rank biserial correlation) of −0.510 (SE = 0.101). The Vovk–Sellke Maximum *p*-Ratio for this test also offered to support the significant difference between the groups.

To investigate the differences between the non-psychiatric (0) and psychiatric (1) groups across various psychosocial measures of BASM-P, we conducted both Student’s *t*-tests and Mann–Whitney U tests. Descriptive statistics, including the mean, standard deviation (SD), standard error (SE), and coefficient of variation (CV) for each measure, are presented below. The results of the statistical tests, including test statistics, *p*-values, and effect sizes, are also provided in Table 3.

The results indicate significant differences between the non-psychiatric and psychiatric groups across all measured variables. For instance, the BASM-P General Factor showed a significant difference with a large effect size. Similar patterns were observed for obsessive thoughts, fear, phobia, maladaptive coping, distress, stress, anxiety, perception of vulnerability to illness, traumatic grief, and grief from job loss, with all comparisons yielding significant *p*-values and notable effect sizes, indicating substantial differences between the groups. These findings suggest that individuals in the psychiatric group exhibit higher levels of distress and maladaptive coping compared to the non-psychiatric group. The consistent significance across both parametric and non-parametric tests further strengthens the internal validity of these results.

### 3.3. Network Analysis of BASM-P Across Diagnosis

Given these differences, it is essential to understand the underlying psychosocial functioning within these groups more comprehensively. To this end, we conducted a network analysis to investigate the relationships between various mental health variables in the non-psychiatric group.

#### 3.3.1. Non-Psychiatric Network Analysis

To investigate the relationships between various mental health variables in the non-psychiatric group, we conducted a network analysis. The network consisted of 10 nodes and 32 non-zero edges out of a possible 45, resulting in a sparsity of 0.289. This analysis revealed significant insights into the interconnections between the variables. The centrality measures, including betweenness, closeness, strength, and expected influence, are presented in Table 4.

Obsessive thoughts emerged as the most central variable in the network, with moderate correlations to fear (*r* = 0.419), distress (*r* = 0.216), and stress (*r* = 0.295). This indicates that obsessive thoughts are a relevant factor in the overall psychosocial functioning profile of the non-psychiatric group. The strongest connection suggests that individuals with high levels of obsessive thoughts also tend to experience high levels of fear. Another connection highlights the association between distress and obsessive thoughts. Stress shows a connection with obsessive thoughts, indicating that higher stress levels are associated with increased obsessive thoughts. Traumatic grief and grief from job loss show connections with other items, but these are weaker compared to the primary connections in the network of Figure 2.

#### 3.3.2. Psychiatric Network Analysis

To explore the relationships among different mental health variables in psychiatric participants, we performed a network analysis. This network comprised 10 nodes and 30 non-zero edges out of a possible 45, yielding a sparsity of 0.333. The analysis revealed significant insights into how these variables are interconnected. Table 5 presents the centrality measures, which include betweenness, closeness, strength, and expected influence.

Obsessive thoughts emerged as a central variable in the psychiatric network, with significant connections to fear (*r* = 0.473), distress (*r* = 0.628), and grief from job loss (*r* = 0.515). This indicates that obsessive thoughts are closely linked to various forms of distress in psychiatric patients. The strongest connection was between obsessive thoughts and fear, suggesting that psychiatric patients with high levels of obsessive thoughts also tend to experience high levels of fear. Another strong connection highlighted the close association between distress and obsessive thoughts, while distress and grief from job loss also showed a higher association.

Moderate connections were observed between stress and maladaptive coping, indicating that high levels of stress are associated with adjustment problems. Obsessive thoughts and distress also showed a moderate connection, further reflecting their significant association. Phobia and perception of vulnerability to illness exhibited a moderate connection, indicating a relationship between perceived vulnerability to illness and phobia. Weaker connections were noted for anxiety, which was linked to several other items but none particularly strongly. This suggests that while anxiety is present in psychiatric patients, it is not as central compared to other factors.

Traumatic grief showed moderate connections with other items, indicating its relevance but not centrality within the network. The analysis also revealed notable negative associations, indicated by red lines in the network plot. For instance, obsessive thoughts showed a negative association with grief from job loss, and fear had a negative association with traumatic grief. These negative associations suggest inverse relationships where an increase in one variable corresponds to a decrease in the other, highlighting complex dynamics within the psychiatric network, as shown in Figure 3.

## 4. Discussion

As mentioned, the objectives of this study were: (1) to evidence the psychometric factorial internal structure of BASM-P in the sample; (2) to investigate the psychosocial impact of the COVID-19 pandemic in Brazilian post-peak period among individuals with and without pre-existing psychiatric conditions using the BASM-P; and (3) to analyze relationships between the mental health variables measured by the BASM-P in both groups. Therefore, it is important to highlight that the BASM-P demonstrated robust psychometric factorial internal structure, with a high KMO value indicating excellent sampling adequacy and a significant Bartlett’s test confirming the suitability of the data for factor analysis. The single-factor solution explained a substantial portion of the variance with reasonable factor loadings, and the internal consistency was high, as indicated by McDonald’s omega. These results validate the BASM-P as a reliable and valid tool for assessing pandemic-related psychosocial impact across different cultural contexts.

BASM-P includes various mental health variables that together provide a comprehensive assessment of an individual’s response to the pandemic. Each variable measures a specific aspect of psychosocial functioning, contributing to the overall BASM-P total score. The total score is the sum of the individual scores for each variable, representing the cumulative level of psychosocial impact experienced by the individual. This approach allows for a detailed examination of mental health and supports our hypotheses regarding the differences between the psychiatric and non-psychiatric groups’ responses in the Brazilian post-peak period.

The results of this study strongly support the first hypothesis. Psychiatric patients exhibited significantly higher scores across various psychosocial measures, including distress, obsessive thoughts, and maladaptive coping. This is evidenced by the substantial differences in BASM-P scores between the two groups, with psychiatric patients showing higher mean scores and greater variability. The significant *t*-test and Mann–Whitney U test results, coupled with large effect sizes, underscore the heightened psychosocial burden experienced by psychiatric patients. These findings align with existing literature, which suggests that individuals with pre-existing psychiatric conditions are more susceptible to increased psychosocial functioning problems during crises such as the COVID-19 pandemic [1,3,40,41,42]. The elevated levels of distress and maladaptive coping observed in psychiatric patients highlight the need for targeted mental health interventions to support this vulnerable population.

The network analysis results provide clear evidence in support of the second hypothesis. In the psychiatric group, obsessive thoughts—usually repetitive, intrusive, unwanted, and disturbing [43]—emerged as a central variable with strong connections to fear, distress, and grief from job loss. Similarly, distress showed significant associations with other mental health variables. These centrality measures indicate that obsessive thoughts and distress are pivotal in the psychiatric network, influencing a wide range of other symptoms and behaviors. The higher centrality values for these variables in the psychiatric group, compared to the non-psychiatric group, suggest a more complex and intertwined psychosocial functioning profile. This complexity reflects the multifaceted nature of psychiatric disorders, where core symptoms such as obsessive thoughts and distress can have extensive ripple effects across other aspects of mental health [44,45]. This finding underscores the importance of focusing therapeutic efforts on these central symptoms to potentially alleviate a broader spectrum of emotional and behavioral problems in psychiatric patients.

The third hypothesis is also supported by the findings of this study. The network analysis revealed a moderate connection between stress and maladaptive coping in psychiatric patients. This relationship was stronger compared to the non-psychiatric group, where the connection between stress and maladaptive coping was less pronounced. This suggests that in psychiatric patients high levels of stress are more likely to lead to adjustment problems, which can further exacerbate their distress. A similar suggestion derives from research developed during the COVID-19 pandemic in Southeastern Mexico [46]. The presence of stronger associations between stress and maladaptive coping in psychiatric patients indicates a process of cyclical nature in this population. Effective stress management interventions are therefore crucial for psychiatric patients to break this cycle and promote healthier coping mechanisms. These findings emphasize the need for comprehensive mental health programs that address both stress reduction and the development of adaptive coping—instrumentalizing emotional and cognitive self-regulation, stimulating information seeking, and encouraging the search for support, for example—to improve overall psychosocial functioning in psychiatric patients.

The network analysis for the non-psychiatric group revealed that obsessive thoughts are a central variable, connected to fear, distress, and stress. This suggests that in individuals without pre-existing psychiatric conditions, obsessive thoughts about illness and the pandemic play a pivotal role in influencing other psychosocial states. The association between obsessive thoughts and fear supports the notion that fear-driven cognitive patterns can lead to heightened distress and stress, potentially due to the pervasive uncertainty and perceived threats posed by the pandemic [47,48]. Connections between stress and maladaptive coping, as well as between stress and fear, indicate that stress management is crucial in non-psychiatric populations to prevent the development of adjustment problems. The weaker connections involving traumatic grief and grief from job loss suggest these factors, while present, are less central in the psychosocial functioning profile of individuals without pre-existing psychiatric conditions during the COVID-19 pandemic.

In the psychiatric group, obsessive thoughts again emerged as a central variable, with significant connections to fear, distress, and grief from job loss. This underscores the intensified impact of obsessive thoughts on psychiatric patients, who already have a heightened baseline of distress [49]. The strong link between obsessive thoughts and fear indicates that interventions targeting obsessive patterns could be particularly beneficial for reducing fear and associated distress in this group. The moderate connections between stress and maladaptive coping strategies in psychiatric patients suggest that stress exacerbates maladaptive behaviors, which are already more prevalent in this population. The presence of negative associations, such as between obsessive thoughts and grief from job loss, points to complex interactions where an increase in one type of distress might correspond to a decrease in another, potentially due to competing cognitive and emotional resources. These negative associations indicate that addressing one symptom could potentially alleviate others, highlighting the importance of integrated and holistic treatment approaches for psychiatric patients.

The network analysis underscores fundamental differences in the psychosocial functioning of psychiatric and non-psychiatric groups. While obsessive thoughts emerged as a central variable in both networks, the strength and pattern of connections to other mental health variables differed significantly. In the non-psychiatric group, obsessive thoughts were primarily associated with fear and stress, reflecting a response pattern rooted in heightened vigilance and perceived threats during the pandemic. This suggests that obsessive thoughts in this group may act as a transient response to the external crisis, modulating emotional states like fear without necessarily leading to broader psychosocial dysfunctions, as indicated by previous research [50,51,52]. In contrast, the psychiatric group exhibited a more intricate network structure, where obsessive thoughts were strongly interconnected with distress and grief from job loss, alongside fear. This points to a deeper entanglement of obsessive thoughts with chronic emotional burdens and maladaptive coping mechanisms in individuals with psychiatric conditions.

The stronger associations observed in the psychiatric network highlight a cumulative effect, where core symptoms like obsessive thoughts exacerbate other psychosocial functioning problems, creating a cycle of heightened distress. Studies on the interplay of symptoms, independent of the COVID-19 context, also support these interpretations [53]. Research on the relationships between obsessive-compulsive symptoms and other mental disorders, such as complex post-traumatic stress disorder, which was recognized as a novel diagnosis in the 11th International Classification of Diseases, highlights the central role of obsessive thoughts within symptom networks [54]. These studies demonstrate how obsessive thoughts often serve as key nodes, strongly influencing other symptoms like distress and maladaptive coping. These findings are convergent with the present study’s results, suggesting that the centrality of obsessive thoughts is not limited to COVID-19 pandemic-related contexts but represents a broader phenomenon in the dynamics of mental disorders.

The results of this study have significant implications for clinical practice and future research. Understanding the heightened levels of distress, obsessive thoughts, and maladaptive coping in psychiatric patients can favor the development of mental health interventions. Focusing on the central variables of obsessive thoughts and distress in therapeutic settings may yield broader improvements in mental health outcomes for psychiatric patients. Additionally, the strong association between stress and maladaptive coping in psychiatric patients suggests that stress management programs should be integrated into treatment plans to enhance coping strategies. Future research should continue to explore the longitudinal impacts of the pandemic on mental health, examining how these psychosocial networks evolve over time. Expanding the sample to include diverse cultural backgrounds will also provide a more comprehensive understanding of how cultural factors influence the psychosocial consequences of global crises. Overall, this study highlights the importance of tailored mental health interventions to address the specific needs of both psychiatric and non-psychiatric populations during and beyond the pandemic.

Understanding the heightened levels of distress, obsessive thoughts, and maladaptive coping in psychiatric patients can inform the development of targeted mental health interventions in post-peak periods of future sanitary crises and in post-COVID-19 pandemic era. For them, interventions should prioritize addressing obsessive thoughts and distress due to their centrality in the psychosocial network. Cognitive-behavioral therapy and mindfulness-based interventions could be particularly effective in reducing these symptoms [55,56]. Dynamic psychotherapy, a development of traditional psychoanalysis, also may produce favorable outcomes, despite not fitting into short-term intervention [57]. Additionally, the strong association between stress and maladaptive coping suggests the need for integrating stress management programs into treatment plans for individuals with pre-existing psychiatric conditions, to enhance adaptive coping strategies, and for individuals without pre-existing psychiatric conditions, to prevent the escalation into more severe distress and maladaptive coping [58,59]. Psychoeducation about healthy coping strategies could be beneficial too [60]. Community mental health centers and telehealth services can be pivotal in delivering these interventions effectively [61,62]. Public health strategies are necessary to incorporate comprehensive stress management programs, particularly in low and middle-income countries.

Future research should focus on longitudinal studies to explore the evolving impacts of the post-COVID-19 pandemic era on mental health. Understanding how psychosocial networks change over time can help public health officials anticipate future mental health needs and adjust strategies accordingly. These studies can provide insights into the long-term effects of the pandemic and inform the development of sustainable mental health interventions. Expanding research to include diverse cultural backgrounds is essential for developing culturally sensitive public health strategies. Cross-cultural studies can reveal how different populations experience and cope with pandemic-related psychosocial impact over time, as indicated by research developed at the beginning of the pandemic [63,64]. This knowledge can help tailor public health interventions to meet the unique needs of various cultural groups, ensuring more effective and inclusive mental health care in the post-COVID-19 pandemic era.

Research should also focus on evaluating the effectiveness of targeted interventions on the central variables identified in this study, particularly in reducing obsessive thoughts and distress in psychiatric patients. These evaluations can inform best practices and guide the implementation of evidence-based public health strategies [5]. Overall, this study underscores the importance of tailored public health interventions to address the specific mental health needs of both psychiatric and non-psychiatric populations during and beyond pandemics or sanitary crises. By focusing on central symptoms and their interconnections, public health professionals can develop more effective strategies to mitigate the psychosocial impact of these events, enhancing the resilience and well-being of affected individuals. The BASM-P may help to design, assess, and evaluate strategies focusing on individuals with and without pre-existing psychiatric conditions. 

There are some limitations to this study that should be acknowledged. First, the cross-sectional design limits the ability to assess causality between the variables analyzed, as the relationships identified represent associations rather than temporal or causal links. This introduces the possibility of reverse causation or unmeasured confounding factors influencing the results. Second, the sample size of individuals with mental disorders was relatively small, potentially affecting the robustness and generalizability of the findings within this subgroup. This limitation could result in reduced statistical power and an increased risk of sampling bias. Third, the disproportion of men and women in the groups may have influenced the results, as gender differences in the psychosocial impact of sanitary crises like the COVID-19 pandemic are well-documented [65,66]. This imbalance could introduce selection bias and limit the applicability of the findings across genders. Additionally, no comparisons were executed between different diagnoses within the psychiatric group, since, as already mentioned, information was not obtained due to ethical restrictions within the health services in which participants in the psychiatric group were being assisted. This restricts the study’s capacity to explore nuanced differences among psychiatric subpopulations. Future research should address these limitations by employing longitudinal designs, larger and more diverse samples, and balanced gender representation to strengthen the validity and applicability of the findings.

## 5. Conclusions

This study described the psychosocial impact of the COVID-19 pandemic in the Brazilian post-peak period among individuals with and without pre-existing psychiatric conditions using the BASM-P, and, therefore, covered a period in which few studies on the topic were developed, especially in Brazil. The relationships between the mental health variables measured by the BASM-P in both groups were also identified, as well as the central factors and their connections in each group. Additionally, a detailed analysis of the psychometric factorial internal structure of the BASM-P was presented here, an instrument that can be useful too in future pandemics or sanitary crises. The results highlight the centrality of obsessive thoughts in shaping the psychosocial impact of the COVID-19 pandemic in the Brazilian post-peak period, with significant differences between the two groups. Psychiatric groups exhibit higher levels of distress, obsessive thoughts, and maladaptive coping strategies compared to the non-psychiatric group. Network analysis further reveals that obsessive thoughts and distress are central to the psychosocial functioning profile of psychiatric patients, suggesting a more complex and intertwined set of relationships between psychosocial variables in this group.

From a public health perspective, this study emphasizes the importance of developing and disseminating targeted mental health strategies to support both individuals with and without pre-existing psychiatric conditions in the post-peak period of future sanitary crises and also in post-COVID-19 pandemic era. Future research should continue to explore the longitudinal impacts of the COVID-19 pandemic on mental health, considering diverse cultural backgrounds to develop culturally sensitive public health strategies. This study advances our understanding of the psychosocial impact of the COVID-19 pandemic and lays the groundwork for future interventions that can significantly impact public health.

## Figures and Tables

**Figure 1 ijerph-22-00027-f001:**
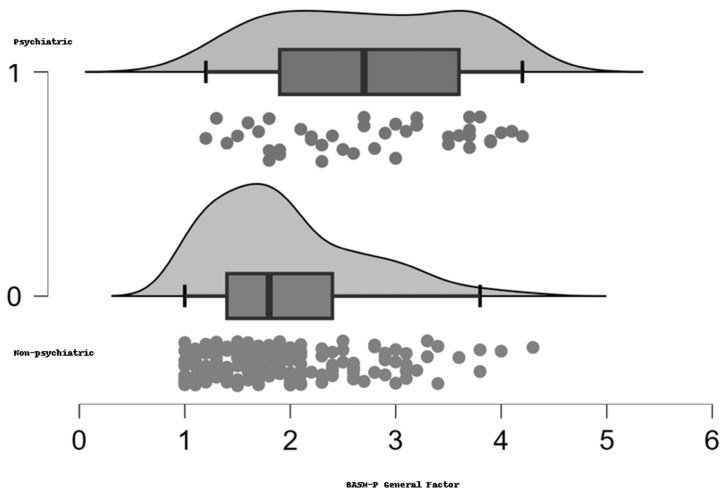
Raincloud distribution for BASM-P across diagnosis.

**Figure 2 ijerph-22-00027-f002:**
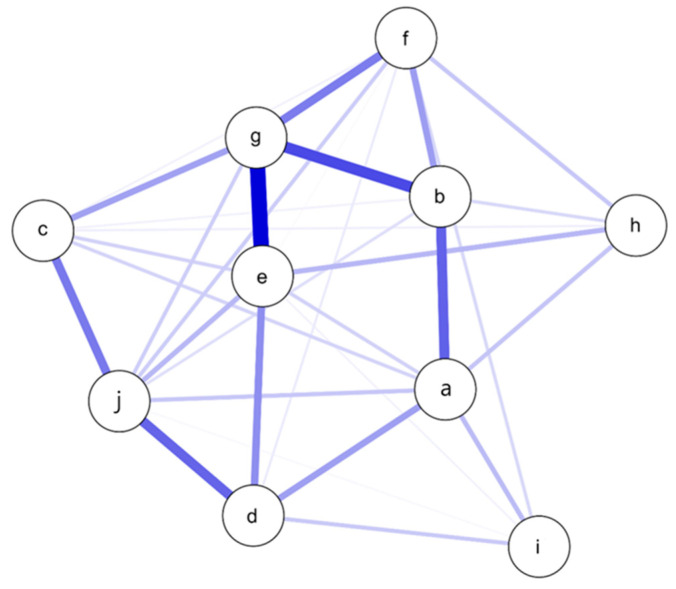
Network plot for the non-psychiatric group ^1^. ^1^ Nodes represent variables: a—phobia, b—stress, c—anxiety, d—vulnerability, e—fear, f—distress, g—obsessive thoughts, h—traumatic grief, i—job loss, j—maladaptive coping. Thicker, darker lines indicate stronger relationships. Blue lines represent positive association.

**Figure 3 ijerph-22-00027-f003:**
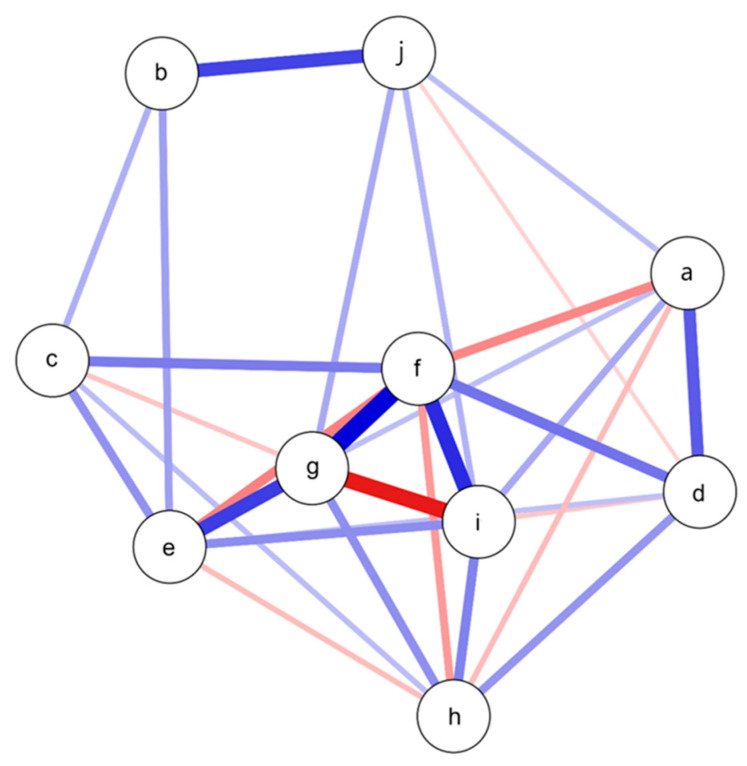
Network plot for the psychiatric group ^1^. ^1^ Nodes represent variables: a—phobia, b—stress, c—anxiety, d—vulnerability, e—fear, f—distress, g—obsessive thoughts, h—traumatic grief, i—job loss, j—maladaptive coping. Thicker, darker lines indicate stronger relationships. Blue lines represent positive association, while red lines represent negative association.

**Table 1 ijerph-22-00027-t001:** Descriptive statistics and Chi-Squared Test results.

Variable		Non-Psychiatric Group	Psychiatric Group	Total Count	Chi-Squared	*p*-Value
Ethnicity ^1^	Non-white	35 (20.83%)	13 (31.71%)	48 (22.97%)	2.203	0.138
	White	133 (79.17%)	28 (68.29%)	161 (77.03%)		
	Total	168 (100%)	41 (100%)	209 (100%)		
Gender	Women	122 (72.62%)	32 (78.05%)	154 (73.68%)	0.501	0.479
	Men	46 (27.38%)	9 (21.95%)	55 (26.32%)		
	Total	168 (100%)	41 (100%)	209 (100%)		
Scholarship	Not graduated	22 (13.10%)	4 (9.76%)	26 (12.44%)	0.337	0.561
	Graduated	146 (86.90%)	37 (90.24%)	183 (87.56%)		
	Total	168 (100%)	41 (100%)	209 (100%)		
Diagnosis of COVID-19	No	99 (58.93%)	26 (63.41%)	125 (59.81%)	0.276	0.599
	Yes	69 (41.07%)	15 (36.59%)	84 (40.19%)		
	Total	168 (100%)	41 (100%)	209 (100%)		
Wage reduction	No	124 (73.81%)	33 (80.49%)	157 (75.12%)	0.786	0.375
	Yes	44 (6.19%)	8 (19.51%)	52 (24.88%)		
	Total	168 (100%)	41 (100%)	209 (100%)		
Family grief during pandemic	No	120 (71.43%)	24 (58.54%)	144 (68.90%)	2.556	0.110
	Yes	48 (28.57%)	17 (41.46%)	65 (31.10%)		
	Total	168 (100%)	41 (100%)	209 (100%)		

^1^ The Brazilian criteria for ethnicity used in this study were based on participants’ self-reports of being white or non-white. It is important to note that this concept of self-report ethnicity may focus on personal comprehension of skin color, which may differ from other cultural understandings of being white and non-white.

**Table 2 ijerph-22-00027-t002:** BASM-P multilingual item wording and factor loading.

Content	Item Wording ^1^				Factor Loading
	Brazilian Portuguese	English	Spanish	French	
Obsessive thoughts	Eu não consegui parar de pensar sobre estar doente.	I couldn’t stop thinking about being ill.	No he podido dejar de pensar en estar enfermo.	Je n’ai pas pu arrêter de penser à être malade.	0.816
Fear	Meu coração acelera ou palpita quando penso que posso adoecer na pandemia.	My heart races or palpitates when I think I might get ill during the pandemic.	Mi corazón se acelera o palpita cuando pienso que puedo enfermarme durante la pandemia.	Mon cœur s’accélère ou palpite quand je pense que je pourrais tomber malade pendant la pandémie.	0.806
Phobia	Tenho sentido dores no peito por conta do medo de adoecer em decorrência da pandemia.	I’ve felt chest pains due to the fear of getting ill from the pandemic.	He sentido dolores en el pecho debido al miedo de enfermarme por la pandemia.	J’ai ressenti des douleurs dans la poitrine à cause de la peur de tomber malade à cause de la pandémie.	0.798
Maladaptive coping	Perdi a esperança de voltar para minha vida normal.	I’ve lost hope of returning to my normal life.	He perdido la esperanza de volver a mi vida normal.	J’ai perdu l’espoir de revenir à ma vie normale.	0.712
Distress	Depois da pandemia, eu frequentemente tenho dor de barriga, gases ou outros desconfortos no estomago.	Since the pandemic, I’ve frequently experienced stomach aches, gas, or other stomach discomforts.	Desde la pandemia, frecuentemente he tenido dolor de estómago, gases u otras molestias estomacales.	Depuis la pandémie, j’ai souvent des maux d’estomac, des gaz ou d’autres inconforts gastriques.	0.679
Stress	Busquei em meu próprio corpo por sinais de infecção relacionados ao adoecimento decorrente da pandemia.	I’ve searched my own body for signs of infection related to illness from the pandemic.	He buscado en mi propio cuerpo signos de infección relacionados con la enfermedad debido a la pandemia.	J’ai cherché dans mon propre corps des signes d’infection liés à la maladie due à la pandémie.	0.675
Anxiety	Perdi apetite quando pensei ou fui exposto a informações sobre a pandemia.	I’ve lost my appetite when I thought about or was exposed to information about the pandemic.	He perdido el apetito cuando pensé o estuve expuesto a información sobre la pandemia.	J’ai perdu l’appétit quand j’ai pensé ou été exposé à des informations sur la pandémie.	0.646
Perception of vulnerability to illness	Meu adoecimento decorrente da pandemia me afeta emocionalmente. (exemplo: sentir-se com raiva, assustado(a), bravo(a) ou deprimido(a)),	My illness due to the pandemic affects me emotionally (e.g., feeling angry, scared, upset, or depressed).	Mi enfermedad debido a la pandemia me afecta emocionalmente (por ejemplo, sentirme enfadado, asustado, molesto o deprimido).	Ma maladie due à la pandémie m’affecte émotionnellement (par exemple, me sentir en colère, effrayé, contrarié ou déprimé).	0.564
Traumatic grief	Eu evitei lugares, objetos ou pensamentos que me lembrassem da pessoa que se foi.	I avoided places, objects, or thoughts that reminded me of the person who passed away.	He evitado lugares, objetos o pensamientos que me recordaban a la persona que se fue.	J’ai évité les lieux, objets ou pensées qui me rappelaient la personne décédée.	0.402
Grief from job loss	Sinto-me culpado(a) pela perda do meu trabalho.	I feel guilty for losing my job.	Me siento culpable por perder mi trabajo.	Je me sens coupable d’avoir perdu mon travail.	0.329

^1^ The current study focuses on the evidence of Brazilian Portuguese item wording. However, we suggest content adaptation for other languages for future studies interested in the transcultural adaptation of BASM-P.

**Table 3 ijerph-22-00027-t003:** Descriptive Statistics and Test Results for Psychosocial Measures of BASM-P by Group ^1^.

Variable	Group	*n*	Mean	SD	SE	CV	*p*
BASM-P General Factor	0	168	1.946	0.711	0.055	0.365	Student: t(206) = −6.101, *p* < 0.001, d = −1.063; t(206) = −6.101, *p* < 0.001, d = −1.063; t(206) = −6.101, *p* < 0.001, d = −1.063 Mann–Whitney: U = 1676.500, *p* < 0.001, *r* = −0.510; U = 1676.500, *p* < 0.001, *r* = −0.510; U = 1676.500, *p* < 0.001, *r* = −0.510
	1	41	2.741	0.887	0.139	0.324	
Obsessive thoughts	0	168	1.470	0.935	0.072	0.636	Student: t(207) = −3.940, *p* < 0.001, d = −0.686; t(207) = −3.940, *p* < 0.001, d = −0.686; t(207) = −3.940, *p* < 0.001, d = −0.686 Mann–Whitney: U = 2385.500, *p* < 0.001, *r* = −0.307; U = 2385.500, *p* < 0.001, *r* = −0.307; U = 2385.500, *p* < 0.001, *r* = −0.307
	1	41	2.171	1.321	0.206	0.609	
Fear	0	168	1.613	1.153	0.089	0.715	Student: t(207) = −4.965, *p* < 0.001, d = −0.865; t(207) = −4.965, *p* < 0.001, d = −0.865; t(207) = −4.965, *p* < 0.001, d = −0.865 Mann–Whitney: U = 2136.500, *p* < 0.001, *r* = −0.380; U = 2136.500, *p* < 0.001, *r* = −0.380; U = 2136.500, *p* < 0.001, *r* = −0.380
	1	41	2.683	1.540	0.241	0.574	
Phobia	0	168	1.470	0.902	0.070	0.613	Student: t(207) = −3.588, *p* < 0.001, d = −0.625; t(207) = −3.588, *p* < 0.001, d = −0.625; t(207) = −3.588, *p* < 0.001, d = −0.625 Mann–Whitney: U = 2345.500, *p* < 0.001, *r* = −0.319; U = 2345.500, *p* < 0.001, *r* = −0.319; U = 2345.500, *p* < 0.001, *r* = −0.319
	1	41	2.073	1.191	0.186	0.575	
Maladaptive coping	0	168	2.108	1.198	0.093	0.568	Student: t(206) = −4.401, *p* < 0.001, d = −0.767; t(206) = −4.401, *p* < 0.001, d = −0.767; t(206) = −4.401, *p* < 0.001, d = −0.767 Mann–Whitney: U = 2061.500, *p* < 0.001, *r* = −0.398; U = 2061.500, *p* < 0.001, *r* = −0.398; U = 2061.500, *p* < 0.001, *r* = −0.398
	1	41	3.049	1.341	0.209	0.440	
Distress	0	168	1.458	0.881	0.068	0.604	Student: t(207) = −4.602, *p* < 0.001, d = −0.802; t(207) = −4.602, *p* < 0.001, d = −0.802; t(207) = −4.602, *p* < 0.001, d = −0.802 Mann–Whitney: U = 2456.000, *p* < 0.001, *r* = −0.287; U = 2456.000, *p* < 0.001, *r* = −0.287; U = 2456.000, *p* < 0.001, *r* = −0.287
	1	41	2.293	1.537	0.240	0.670	
Stress	0	168	2.101	1.321	0.102	0.629	Student: t(207) = −3.625, *p* < 0.001, d = −0.631; t(207) = −3.625, *p* < 0.001, d = −0.631; t(207) = −3.625, *p* < 0.001, d = −0.631 Mann–Whitney: U = 2241.500, *p* < 0.001, *r* = −0.349; U = 2241.500, *p* < 0.001, *r* = −0.349; U = 2241.500, *p* < 0.001, *r* = −0.349
	1	41	2.951	1.448	0.226	0.491	
Anxiety	0	168	1.232	0.599	0.046	0.486	Student: t(207) = −4.295, *p* < 0.001, d = −0.748; t(207) = −4.295, *p* < 0.001, d = −0.748; t(207) = −4.295, *p* < 0.001, d = −0.748 Mann–Whitney: U = 2453.500, *p* < 0.001, *r* = −0.288; U = 2453.500, *p* < 0.001, *r* = −0.288; U = 2453.500, *p* < 0.001, *r* = −0.288
	1	41	1.756	1.019	0.159	0.580	
Perception of vulnerability to illness	0	168	3.310	1.340	0.103	0.405	Student: t(207) = −2.474, *p* = 0.014, d = −0.431; t(207) = −2.474, *p* = 0.014, d = −0.431; t(207) = −2.474, *p* = 0.014, d = −0.431 Mann–Whitney: U = 2611.500, *p* = 0.014, *r* = −0.242; U = 2611.500, *p* = 0.014, *r* = −0.242; U = 2611.500, *p* = 0.014, *r* = −0.242
	1	41	3.878	1.229	0.192	0.317	
Traumatic grief	0	168	1.506	0.841	0.065	0.558	Student: t(207) = −4.147, *p* < 0.001, d = −0.722; t(207) = −4.147, *p* < 0.001, d = −0.722; t(207) = −4.147, *p* < 0.001, d = −0.722 Mann–Whitney: U = 2528.500, *p* = 0.002, *r* = −0.266; U = 2528.500, *p* = 0.002, *r* = −0.266; U = 2528.500, *p* = 0.002, *r* = −0.266
	1	41	2.195	1.327	0.207	0.605	
Grief from job loss	0	168	1.964	1.276	0.098	0.649	Student: t(207) = −3.989, *p* < 0.001, d = −0.695; t(207) = −3.989, *p* < 0.001, d = −0.695; t(207) = −3.989, *p* < 0.001, d = −0.695 Mann–Whitney: U = 2315.000, *p* < 0.001, *r* = −0.328; U = 2315.000, *p* < 0.001, *r* = −0.328; U = 2315.000, *p* < 0.001, *r* = −0.328
	1	41	2.902	1.625	0.254	0.560	

^1^ Group 0 = Non-psychiatric; Group 1 = Psychiatric; CV = Coefficient of Variation; For the Student *t*-test, effect size is given by Cohen’s d. For the Mann–Whitney test, effect size is given by the rank biserial correlation.

**Table 4 ijerph-22-00027-t004:** Centrality Measures of the Non-Psychiatric Network.

Variable	Betweenness	Closeness	Strength	Expected Influence
Obsessive thoughts	1.776	1.330	1.524	1.524
Fear	1.230	1.097	1.000	1.000
Phobia	0.410	0.506	0.424	0.424
Maladaptive coping	−0.410	−0.391	0.607	0.607
Distress	−0.957	−0.044	−0.260	−0.260
Stress	0.410	0.511	0.390	0.390
Anxiety	−0.957	−0.556	−0.582	−0.582
Perception of vulnerability to illness	0.410	0.589	−0.109	−0.109
Traumatic grief	−0.957	−1.334	−1.305	−1.305
Grief from job loss	−0.957	−1.708	−1.689	−1.689

**Table 5 ijerph-22-00027-t005:** Centrality Measures of the Psychiatric Network.

Variable	Betweenness	Closeness	Strength	Expected Influence
Obsessive thoughts	0.543	1.454	1.278	1.123
Fear	0.543	0.320	0.358	0.589
Phobia	−0.416	−0.458	−0.443	−1.667
Maladaptive coping	−0.096	−0.945	−0.897	0.463
Distress	2.461	1.540	1.643	0.644
Stress	−0.096	−1.351	−1.297	0.335
Anxiety	−0.735	−0.302	−0.947	−0.045
Perception of vulnerability to illness	−0.735	−0.510	−0.429	0.486
Traumatic grief	−0.735	−0.615	−0.136	−1.907
Grief from job loss	−0.735	0.868	0.870	−0.021

## Data Availability

The original contributions presented in this study are included in the article. Further inquiries can be directed to the corresponding author.

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
