# Peer review of "Psychosocial Impact of the COVID-19 Pandemic in Brazilian Post-Peak Period: Differences Between Individuals with and Without Pre-Existing Psychiatric Conditions"

_ijerph, 2024, doi:10.3390/ijerph22010027_

Round 1

Reviewer 1 Report

Comments and Suggestions for Authors

Dear Editor,

This is an interesting manuscript, in which the authors evaluated the psychosocial impacts of the Covid-19 Pandemic among individuals with and without psychiatric disorders prior to the health crisis. Although the objective of the study is quite relevant, there are several methodological limitations that reduce the importance of the results. I suggest, therefore, some structural changes in order to make the article more attractive and potentially publishable.

1.       The authors presented evidence of the validity of an instrument that assesses mental health – pandemic version, for the Brazilian population, and found important psychometric properties. However, they did not present the validation process as the objective of the study (this does not appear in the abstract, for example, and appears as a secondary objective at the end of the introduction. My suggestion is that you restructure the article and focus on the validation process of the instrument, because almost all of the introduction is related to the validated instrument, and little has been explored in terms of distress, obsessive thoughts, and maladaptive coping strategies.

2.       I also suggest that they add a paragraph to justify the study, at the end of the introduction, emphasizing the importance of validating and investigating psychosocial variables in health crises. The justification for the study should focus on public health issues, since the authors chose a journal in this area. What needs to be answered here is: how does the validation of an instrument for the assessment of cognitive and behavioral problems contribute to the Brazilian public health system, highlighting the differences of this instrument from others already validated for the same purpose. Subsequently, the authors should also justify the importance of evaluating the variables measured by the BASM-P in the Brazilian population.

3.       This is a cross-sectional study, in which participants were selected by convenience, and the sample was non-probabilistic. This information should be included in the study method.

4.       The conclusion of the abstract is generic. I suggest you rewrite it.

5.       Line 56, please change "principal" to "main"

6.       In the group of people with mental disorders, how was the diagnosis made? What were the most frequent diagnoses?

7.       In table 1, the authors should add the percent values, since we are comparing groups that are numerically very unequal. This would facilitate comparisons.

8.       Also in table 1, it is clear that the groups are homogeneous in terms of sociodemographic variables. This is good, and it needs to be explicit in the results.

9.       In table 1, I suggest removing the column from df.

10.   In the characterization of the BASM-P, in the study method, the authors should indicate whether or not there is a cut-off point for the classification of individuals with or without mental health problems.

11.   I did not find the inclusion and exclusion criteria of the participants in the method. What were these criteria?

12.   How did the authors ensure that the comparison group (without psychiatric disorders, in fact, did not have a psychiatric disorder? Was there an evaluation of these participants by a qualified mental health professional, for the inclusion of the subjects in the research?

13.   It is necessary, in the method, to add a “Setting” section, and explain where the study was carried out – in which region of the country, city and institution, and briefly characterize them.

14.   In line 124, I suggest that you add the values of Cronbach's alpha and McDonald's Omega.

15.   In item 3.2 (page 7, line 201), I suggest that the authors first present all the descriptive results and then the results of inferential analyses (please reverse the order of the paragraphs).

16.   In line 265, the authors presented spearman's r of 0.216 as a strong correlation, which is not correct – I suggest correction.

17.   From what I see, the results of the network analyses were quite similar in the group diagnosed with a mental disorder and undiagnosed individuals, with obsessive thoughts emerging as a central point – I suggest that the authors deepen the discussion of this result.

18.   The secondary objective should be the one presented in lines 324, 325 and 326, which differs from what was presented at the end of the introduction.

19.   I suggest that you remove what is presented in line 350, since the authors cannot assess causality between the pandemic and the worsening of mental health problems in a cross-sectional study. I also suggest that you remove what is written in lines 322 and 323, for the same reason as above.

20.   In the discussion, I suggest that the authors present a definition of "obsessive thoughts", a central point of network analysis.

21.   I suggest that you delve deeper into the point of adaptive coping, with some examples, and how this concept could contribute to facing new health crises.

22.   Still in the discussion, I suggest that you compare the results of the network analysis of this study with similar studies, indicating main similarities and differences.

23.   In line 425, the authors refer to Cognitive and Behavioral Therapies as techniques, which is incorrect and requires correction.

24.   The authors should avoid generalizations of the results found, since the sample evaluated was quite small, and not representative. For example, I suggest smoothing out the sentence on line 423.

25.   Please remove the paragraph about limitations of the study from the conclusion, and present it at the end of the discussion section.

26.   I suggest that they add as a limitation of the study the cross-sectional design, which makes it impossible to assess causality between the variables analyzed, the small sample size with mental disorders, and the disproportion of men and women in the groups.

27.   Please indicate the possible biases related to each of the limitations.

Best Regards,

Reviwer

Comments on the Quality of English Language

The article is relatively well written, however, some expressions are literally translated from Portuguese to English, and need to be reformulated.

Author Response

Comment 1: “The authors presented evidence of the validity of an instrument that assesses mental health – pandemic version, for the Brazilian population, and found important psychometric properties. However, they did not present the validation process as the objective of the study (this does not appear in the abstract, for example, and appears as a secondary objective at the end of the introduction. My suggestion is that you restructure the article and focus on the validation process of the instrument, because almost all of the introduction is related to the validated instrument, and little has been explored in terms of distress, obsessive thoughts, and maladaptive coping strategies”.

Response 1: Thank you for point this out. We agree with this comment. We have, accordingly, redefined the objectives in the abstract, in the “Introduction” section (p. 2, last paragraph in the reformulated version of the article) and in the “Discussion” section (p. 2, last paragraph in the reformulated version of the article). We have, also, expanded the “Introduction” section to not limit the content to the validation process of instruments and to provide an overview of the psychosocial impact of the COVID-19 pandemic in Brazil (page 2, paragraphs 3 and 4 in the reformulated version of the article), according to the comment 1 presented by the reviewer 3, and to emphasize the relevance of investigate psychosocial variables in health crises (page 3, paragraphs 1 and 2 in the reformulated version of the article), according to the comment 2 presented by the reviewer 1.

Comment 2: “I also suggest that they add a paragraph to justify the study, at the end of the introduction, emphasizing the importance of validating and investigating psychosocial variables in health crises. The justification for the study should focus on public health issues, since the authors chose a journal in this area. What needs to be answered here is: how does the validation of an instrument for the assessment of cognitive and behavioral problems contribute to the Brazilian public health system, highlighting the differences of this instrument from others already validated for the same purpose. Subsequently, the authors should also justify the importance of evaluating the variables measured by the BASM-P in the Brazilian population”.

Response 2: Thank you for point this out. We agree with this comment. We have, accordingly, added two paragraphs (page 3, paragraphs 1 and 2 in the reformulated version of the article) on the suggested content, as mentioned above.

Comment 3: “This is a cross-sectional study, in which participants were selected by convenience, and the sample was non-probabilistic. This information should be included in the study method”.

Response 3: Thank you for point this out. We agree with this comment. We have, accordingly, added the suggested information in “Method” section (page 3, “Study design” sub-section) and in the abstract.

Comment 4: “The conclusion of the abstract is generic. I suggest you rewrite it”.

Response 4: Thank you for point this out. We agree with this comment. We have, accordingly, reformulated the conclusion of the abstract (last sentence).

Comment 5: “Line 56, please change "principal" to "main"”.

Response 5: Thank you for point this out. We agree with this comment. We have, accordingly, changed the mentioned word (p. 2, line 11, in the reformulated version of the article).

Comment 6: “In the group of people with mental disorders, how was the diagnosis made? What were the most frequent diagnoses?”.

Response 6: Thank you for point this out. We agree with the importance of clarify this questions. We have, accordingly, provided information about it on the “Participants” sub-section (p. 3, paragraph 5 in the reformulated version of the article).

Comment 7: “In table 1, the authors should add the percent values, since we are comparing groups that are numerically very unequal. This would facilitate comparisons”.

Response 7: Thank you for point this out. We agree with this comment. We have, accordingly, added percent values in Table 1, and in the “Participants” sub-section (p. 3, last paragraph, and p. 4, first paragraph in the reformulated version of the article).

Comment 8: “Also in table 1, it is clear that the groups are homogeneous in terms of sociodemographic variables. This is good, and it needs to be explicit in the results”.

Response 8: Thank you for point this out. We agree with this comment. We have, accordingly, emphasized the homogeneity of the samples in terms of sociodemographic variables in the “Results” section (p. 6, paragraph 4 in the reformulated version of the article).

Comment 9: “In table 1, I suggest removing the column from df.”.

Response 9: Thank you for point this out. We agree with this comment. We have, accordingly, removed the column from df. In Table 1.

Comment 10: “In the characterization of the BASM-P, in the study method, the authors should indicate whether or not there is a cut-off point for the classification of individuals with or without mental health problems”.

Response 10: Thank you for point this out. We agree with the importance of clarify this questions. We have, accordingly, provided information about it in the “Instruments” sub-section (p. 5, paragraph 3 in the reformulated version of the article).

Comment 11: “I did not find the inclusion and exclusion criteria of the participants in the method. What were these criteria?”.

Response 11: Thank you for point this out. We agree with the importance of clarify this questions. We have, accordingly, provided information about it in the “Participants” sub-section (p. 3, paragraphs 4 and 5 in the reformulated version of the article).

Comment 12: “How did the authors ensure that the comparison group (without psychiatric disorders, in fact, did not have a psychiatric disorder? Was there an evaluation of these participants by a qualified mental health professional, for the inclusion of the subjects in the research?”.

Response 12: Thank you for point this out. We agree with the importance of clarify this questions. We have, accordingly, provided information about it in the “Participants” sub-section (p. 3, paragraph 5 in the reformulated version of the article).

Comment 13: It is necessary, in the method, to add a “Setting” section, and explain where the study was carried out – in which region of the country, city and institution, and briefly characterize them”.

Response 13: Thank you for point this out. We agree with this comment. We have, accordingly, added a “Setting” sub-section with the requested information (p. 4, paragraph 2 in the reformulated version of the article).

Comment 14: “In line 124, I suggest that you add the values of Cronbach's alpha and McDonald's Omega”.

Response 14: Thank you for point this out. We agree with this comment. We have, accordingly, added the requested values (p. 5, paragraph 2, in the reformulated version of the article).

Comment 15: “In item 3.2 (page 7, line 201), I suggest that the authors first present all the descriptive results and then the results of inferential analyses (please reverse the order of the paragraphs)”.

Response 15: Thank you for point this out. We agree with this comment. We have, accordingly, changed the sequence of the referred results (p. 8, last paragraph 2, p. 9, first paragraph in the reformulated version of the article).

Comment 16: “In line 265, the authors presented spearman's r of 0.216 as a strong correlation, which is not correct – I suggest correction”.

Response 16: Thank you for point this out. We agree with this comment. We have, accordingly, corrected the information (p. 11, last paragraph in the reformulated version of the article).

Comment 17: “From what I see, the results of the network analyses were quite similar in the group diagnosed with a mental disorder and undiagnosed individuals, with obsessive thoughts emerging as a central point – I suggest that the authors deepen the discussion of this result”.

Response 17: Thank you for point this out. We agree with this comment. We have, accordingly, deepened the discussion of the referred result (p. 15, paragraphs 2 and 3 in the reformulated version of the article).

Comment 18: “The secondary objective should be the one presented in lines 324, 325 and 326, which differs from what was presented at the end of the introduction”.

Response 18: Thank you for point this out. We agree with this comment. We have, accordingly, padronized the objectives in the abstract, in the “Introduction” section (p. 2, last paragraph in the reformulated version of the article) and in the “Discussion” section (p. 2, last paragraph in the reformulated version of the article).

Comment 19: “I suggest that you remove what is presented in line 350, since the authors cannot assess causality between the pandemic and the worsening of mental health problems in a cross-sectional study. I also suggest that you remove what is written in lines 322 and 323, for the same reason as above”.

Response 19: Thank you for point this out. We agree with this comment. We have, accordingly, mentioned the impossibility of accessing causality in a cross-sectional study in the “Discussion” section (p. 16, last paragraph in the reformulated version of the article).

Comment 20: “In the discussion, I suggest that the authors present a definition of "obsessive thoughts", a central point of network analysis”.

Response 20: Thank you for point this out. We agree with this comment. We have, accordingly, added the requested content in “Discussion” section (p. 14, paragraph 3 in the reformulated version of the article).

Comment 21: “I suggest that you delve deeper into the point of adaptive coping, with some examples, and how this concept could contribute to facing new health crises”.

Response 21: Thank you for point this out. We agree with this comment. We have, accordingly, added some examples of adaptative coping in “Discussion” section (p. 14, paragraph 4 in the reformulated version of the article).

Comment 22: “Still in the discussion, I suggest that you compare the results of the network analysis of this study with similar studies, indicating main similarities and differences”.

Response 22: Thank you for point this out. We agree with this comment. We have, accordingly, developed the request comparisons in “Discussion” section (p. 15, paragraphs 2 and 3 in the reformulated version of the article).

Comment 23: “In line 425, the authors refer to Cognitive and Behavioral Therapies as techniques, which is incorrect and requires correction”.

Response 23: Thank you for point this out. We agree with this comment. We have, accordingly, eliminated the incorrect information in “Discussion” section (p. 16, line 12 in the reformulated version of the article).

Comment 24: “The authors should avoid generalizations of the results found, since the sample evaluated was quite small, and not representative. For example, I suggest smoothing out the sentence on line 423”.

Response 24: Thank you for point this out. We agree with this comment. We have, accordingly, reformulated the referred sentence in “Discussion” section (p. 16, paragraph 1 in the reformulated version of the article).

Comment 25: “Please remove the paragraph about limitations of the study from the conclusion, and present it at the end of the discussion section”.

Response 25: Thank you for point this out. We agree with this comment. We have, accordingly, changed the positioning of the referred content to the “Discussion” section (p. 16, last paragraph in the reformulated version of the article).

Comment 26: “I suggest that they add as a limitation of the study the cross-sectional design, which makes it impossible to assess causality between the variables analyzed, the small sample size with mental disorders, and the disproportion of men and women in the groups”.

Response 26: Thank you for point this out. We agree with this comment. We have, accordingly, added the referred information in the “Discussion” section (p. 16, last paragraph in the reformulated version of the article).

Comment 27: “Please indicate the possible biases related to each of the limitations”.

Response 27: Thank you for point this out. We agree with this comment. We have, accordingly, added the referred information in the “Discussion” section (p. 16, last paragraph in the reformulated version of the article).

Reviewer 2 Report

Comments and Suggestions for Authors

First of all, I would like to congratulate you on your work. It offers an important step forward in favour of public health. However, the following modifications are necessary to improve it. In particular, they would be the following:

- Both in the summary and in the results section, the statistics (for example, p, r, t, d, should always be in italics).

- Section 2.1. The SD statistics should be in italics. In Table 1, the letter p (statistic) should be in italics.

- Section 3.1. The p should also be in italics.

- Section 3.2. The SD, t, p (in italics). On the other hand, in Table 3, the results obtained in p should be given as an example as follows:

Student: t(206)=−6.101; p<.001, d=−1.063; t(206) = -6.101, p < .001, d = -1.063; t(206)=−6.101, p<.001, d=−1.063

Mann-Whitney: U=1676.500, p<.001, r=−0.510; U = 1676.500; p < .001, r = -0.510; U=1676.500, p<.001, r=−0.510

- References: They should be changed to Vancouver format.

Author Response

Comment 1: “Both in the summary and in the results section, the statistics (for example, p, r, t, d, should always be in italics)”.

Response 1: Thank you for point this out. We agree with this comment. Now the statistics are in italics in the abstract, in the “Results” section (p. 6-13, in the reformulated version of the article) and throughout the entire text.

Comment 2: “Section 2.1. The SD statistics should be in italics. In Table 1, the letter p (statistic) should be in italics”.

Response 2: Thank you for point this out. We agree with this comment. Now the SD statistics and the letter p are in italics in Table 1 throughout the entire text.

Comment 3: “Section 3.1. The p should also be in italics”.

Response 3: Thank you for point this out. We agree with this comment. Now the p is in italics throughout the entire text.

Comment 4: “Section 3.2. The SD, t, p (in italics). On the other hand, in Table 3, the results obtained in p should be given as an example as follows: Student: t(206)=−6.101; p<.001, d=−1.063; t(206) = -6.101, p < .001, d = -1.063; t(206)=−6.101, p<.001, d=−1.063

Mann-Whitney: U=1676.500, p<.001, r=−0.510; U = 1676.500; p < .001, r = -0.510; U=1676.500, p<.001, r=−0.510”

Response 4: Thank you for point this out. We agree with this comment. Now the results in Table 3 are presented according to the referred suggestion.

Comment 5: “References: They should be changed to Vancouver format”.

Response 5: The references are presented according to the IJERPH “Instructions for authors” (https://www.mdpi.com/journal/ijerph/instructions), and that's why Vancouver format wasn't exactly followed.

Reviewer 3 Report

Comments and Suggestions for Authors

The paper is examining the psychosocial effects of the recent pandemic in individuals with and without registered psychiatric conditions in Brazil. While the research goals are interesting, I hope that my comments can improve the current version of the paper:

1) A detailed literature review part is missing.  In my opinion, it is important to underline the up-to-date findings on the psychosocial effects of pandemic with special focus on the COVID pandemic, in order to further highlight the added research value of the paper. What other studies find regarding stress, anxiety, fear and the like? How is the pandemic found to affect individuals with existing mental conditions? Are there related available studies for Brazil? These questions should be addressed in a more detailed literature review analysis in my opinion.

2) While the empirical framework is sufficient in my opinion, still there are shortcomings that should be addressed. In the conclusions section the limitations of the study such as the low number of participants, the difficulty to obtain information about health care services demand and the way they might mediate the relationships of interest, etc. should be discussed.

Author Response

Comment 1: “A detailed literature review part is missing.  In my opinion, it is important to underline the up-to-date findings on the psychosocial effects of pandemic with special focus on the COVID pandemic, in order to further highlight the added research value of the paper. What other studies find regarding stress, anxiety, fear and the like? How is the pandemic found to affect individuals with existing mental conditions? Are there related available studies for Brazil? These questions should be addressed in a more detailed literature review analysis in my opinion”

Response 1: Thank you for point this out. We agree with this comment. We have, accordingly, added to the “Introduction” section an overview of the psychosocial impact of the COVID-19 pandemic in Brazil (page 2, paragraphs 3 and 4 in the reformulated version of the article). Also to improve the “Introduction” section, we have added, according to the suggestion presented by the reviewer 1, content to emphasize the relevance of investigate psychosocial variables in health crises (page 3, paragraphs 1 and 2 in the reformulated version of the article).

Comment 2: “While the empirical framework is sufficient in my opinion, still there are shortcomings that should be addressed. In the conclusions section the limitations of the study such as the low number of participants, the difficulty to obtain information about health care services demand and the way they might mediate the relationships of interest, etc. should be discussed”.

Response 2: Thank you for point this out. We agree with this comment. We have, accordingly, added the requested information, and also changed the positioning of the limitations content to the “Discussion” section (p. 16, last paragraph, and p. 17, first paragraph in the reformulated version of the article), according to the suggestion presented by the reviewer 1.

Round 2

Reviewer 1 Report

Comments and Suggestions for Authors

Dear Authors, 

Congratulations on the work done in reformulating the article. My suggestion is that it be published after minor revisions to the English, especially the paragraphs added in this latest version presented. Best regards.